# Study on the Effect of Salt Solution on Durability of Basalt-Fiber-Reinforced Polymer Joints in High-Temperature Environment

**DOI:** 10.3390/polym14112250

**Published:** 2022-05-31

**Authors:** Yisa Fan, Xiaopeng Wang, Ye Liu, Zhen Liu, Gaolei Xi, Linjian Shangguan

**Affiliations:** 1School of Mechanical Engineering, North China University of Water Resources and Electric Power, Zhengzhou 450045, China; fanyisa123@163.com (Y.F.); 15038405185@163.com (X.W.); liuye1524@163.com (Y.L.); lz17335585447@163.com (Z.L.); 2International Joint Laboratory of Thermo-Fluid Electro-Chemical System for New Energy Vehicle, Zhengzhou 450000, China; 3Technology Center for China Tobacco Henan Industrial Limited Company, Zhengzhou 450000, China

**Keywords:** adhesive bonding, BFRP, salt solution, failure mechanism, durability

## Abstract

Due to the low price and good comprehensive properties, FRP composite material has become a new type of civil application material in recent years. In this paper, Araldite^®^ 2012 adhesive was used to bond basalt-fiber-reinforced polymer (BFRP), and the durability of its bonded joints was investigated. Experiments were carried out at 80 °C/DI water (deionized water), 80 °C/3.5% NaCl solution (3.5% SS), and 80 °C/5.0% NaCl solution (5.0% SS) at 0- (unaged), 10-, 20-, and 30-day aging. The specimen and BFRP in the environment of 80 °C/DI water, 80 °C/3.5% SS, and 80 °C/5.0% SS found salt solution under the condition of all sample water absorption decreases, and the activity of salt solution chemistry was weaker compared with deionized water. The load–displacement curve of the joint failure was obtained through quasi-static tensile experiments, and it was found that the adhesive would undergo a post-curing reaction that had a positive impact on the stiffness of the joint in a high-temperature environment. At the same time, it was found that the joint failure strength decreased less in the salt solution environment, and deionized water was more destructive than the salt solution. Referring to the change in water absorption, it was found that the change in the mechanical properties of the joint was mainly related to the permeation effect of the polymer. The change in the T_g_ of adhesive was measured by differential scanning calorimetry (DSC). It was found that T_g_ would decrease after aging, and the change in T_g_ was mainly related to the mobility of the molecular chain. Thermogravimetric analysis (TGA) was used to analyze the thermal behavior of the epoxy resin and some organic matter, and the main weight loss stage was 340–450 °C, which was the complete degradation of epoxy resin and some organic matter. Macro visual and microscopic scanning electron microscope (SEM) and energy dispersive X-ray spectroscopy (EDX) were used to analyze the failure section, and it can be concluded that the failure mode of joint tear failure transitioned to cohesion in the late–mixed interface failure, at the visible interface between the fiber and the resin matrix.

## 1. Introduction

With the rapid growth of global car ownership, environmental pollution, the energy shortage, and other problems are becoming more serious, and energy conservation and environmental protection have become the current development trends of the global automobile industry. New environmentally friendly energy vehicles with less energy consumption and less environmental pollution will eventually replace fuel vehicles, which will become the general trend of the automotive industry. In the research and development of new energy vehicles, low weight is one of the most effective measures to reduce energy consumption and emissions [1]. Automotive weight is mainly optimized through three aspects of body structure, material, and process to improve the strength of the vehicle and reduce the vehicle’s spare quality [2,3]. In lightweight automobile bodies, the thickness is reduced, the car body structure is optimized, and automobile manufacturers have begun to use composite materials and adhesive connection for the body parts to improve the local stiffness of the car body, reduce noise, and improve the purpose of the resonance [4]. Traditional connection methods, such as riveting, bolting, threading, etc., are not suitable for the connection of FRP materials [5]. These methods often result in wall damage, fiber tearing, and unavoidable stress concentrations. Compared with traditional joining methods, the use of adhesive connections has the advantages of lower manufacturing costs, reduced weight, good resistance to static and dynamic loads, improved damage tolerance, and quasi-uniform stress distribution. FRP composite materials have the characteristics of corrosion resistance, high strength-to-weight ratio, flexible design ability, high vibration performance, and impact resistance, which are suitable for civil infrastructure [6] materials. Commonly used FRPs are glass-fiber-reinforced polymer (GFRP), carbon-fiber-reinforced polymer (CFRP), basalt-fiber-reinforced polymer (BFRP), and aramid-fiber-reinforced polymer (AFRP). Most of these FRP materials are fabricated through a pultrusion process [7]. Each material has different properties, advantages, and limitations, and the most suitable FRP material is usually selected based on the type of load, environmental conditions, and cost. CFRP has excellent mechanical properties, fatigue resistance, and corrosion resistance; however, the high cost of carbon fiber limits its application in engineering. GFRP is relatively inexpensive; however, long-term exposure to complex service environments, especially to alkaline environments, make GFRP susceptible to chemical degradation [8]. In addition to the conventional characteristics of low weight, high strength, good ductility, high temperature resistance, corrosion resistance, etc., CFRP generally has superior mechanical properties and chemical stability than GFRP, which is more economical than CFRP [9,10]. Zhang [11] and Elmahdy et al. [12] conducted high-strain rate tensile tests on basalt-reinforced composites, and the data from both experiments verified the abovementioned advantages. It is proved that basalt-fiber-reinforced resin matrix (BFRP) as a new kind of composite material has very broad prospects in machinery manufacturing, the aviation industry, and the automobile industry [6,13]. 

In the manufacturing process of automobile bodies, many materials need to be joined, and adhesive joints are widely used. Galvez et al. [14] proposed a new concept of CFRP-bonded joints in passenger car steel structures. Bonded joints usually ensure adequate mechanical properties over a long period of time. However, during long-term use, bonded joints can be subjected to harsh environmental conditions such as extreme temperatures, humidity, and UV exposure, where continuous damp heat exposure is often the main threat. In terms of temperature studies, S. Abdel-Monsef et al. [15] observed that extremely long aging time had a significant effect on the fracture response of bonded joints, while increasing temperature had a slight effect. Long-term and even short-term exposure to high temperatures change the chemical and physical properties of the adhesive, and the bond strength changes [16]. Both durability and thermal fatigue have adverse effects on mechanical properties of adhesive joints [17]. Tara Sen et al. [18] investigated the flexural strength of reinforced concrete beams externally bonded to isofluid-, glass-fiber-, and carbon-fiber-reinforced composites. It was found that the confinement strength of the isocube FRP fully restrained cylinder strength was close to the GFRP restraint strength, and the mechanical properties of natural fibers were enhanced after heat treatment. Yao et al. [19] analyzed the influence of temperature on BFRP–steel single-lap joints, and they found that the average failure strength increased in the range of −25–50 °C, but decreased significantly in the range of 50–100 °C. Shear stiffness increases in the range of −25–25 °C, but decreases in the range of 25–100 °C. Thus, they concluded that high temperatures have a significant effect on mechanics performance. In addition, along with the increasing use of adhesives, manufacturers and designers in the aerospace, automotive, marine, and construction industries have discovered new possible uses. Due to the newly developed structural adhesive, high-strength applications are possible [20]. Adhesives have proved to be a suitable material for structural applications, and along these lines, adhesives can be found in many different industrial fields. The use of structural adhesives is undoubtedly the most common and effective technique for preventing structural degradation [21]. 

In terms of the influence of moisture on bonding joints, the existence of moisture has been proved to be a factor leading to significant strength degradation of bonding joints [22]. Water can enter the bonding zone through adhesive diffusion, interfacial transport, capillary action, and diffusion through composites [23], affecting the resin matrix and adhesive, leading to expansion, plasticization, cracking, and hydrolysis [24]. Destructive effects of salt solutions on the interlaminar strength of FRPs compared with pure water have been previously reported by someone researchers, usually attributed to resin/fiber interface damage [25]. Zhou and Lucas [26] studied the interaction between water molecules and epoxy resin. The results show that two kinds of bound water can be found in epoxy resin. Type I bound water is shown to destroy weaker interchain van der Waals forces and act as a plasticizer. Type II bound water is the product of strong hydrogen bonds between water molecules and resin networks. Type I bound water is more difficult to remove and causes irreversible material changes, but it increases with higher temperatures and longer exposure time. Heshmati et al. [27] studied the effect of moisture on the mechanical properties of epoxy adhesives and found a direct relationship between moisture content and mechanical properties. Soles et al. [28] proved the relationship between adhesive surface topology and high hygroscopic sensitivity, while Rider et al. [29] linked this high hygroscopic sensitivity to the polarity of the resin. Lu et al. [30] studied the effects of thermal aging, water immersion, and alkali immersion on the mechanical properties of BFRP plates, and the results showed that, in water or alkaline solution, the tensile strength and interlaminar shear strength of thermal aging BFRP plate samples decreased significantly, while the tensile modulus decreased only slightly. The higher the heat aging and immersion temperature, the more severe the observed degradation in tensile strength and interlaminar shear strength.

Temperature and humidity are two distinct effects on bonded joints, and exposure to a combination of humidity and temperature can cause more damage than individual conditions [31]. Heshmati M et al. [32] conducted a comprehensive review of the environmental durability of bonded FRP/steel joints in civil engineering applications and determined that the combined influence of moisture and temperature is the most critical environmental factor. Work on the effects of humidity and temperature on the mechanical properties of adhesive joints using epoxy resins can be found in the literature [10,33]. Viana et al. [34] published a review on temperature and humidity degradation of adhesive joints, and studied the influence of harsh environment on shear tensile strength of multi-material joints. S. Aria et al. [35] studied the mechanical behavior of adhesive joints in different humid and hot environments, such as seawater or distilled water immersion, and proved the applicability of adhesives in humid and hot environments. Sheng Li et al. [36] found that the degradation degree of FRP composites increases with the increase in exposure time, temperature, stress level, and alkalinity/salinity. The impact strength, Young’s modulus, and fatigue life of glass-/basalt-fiber-reinforced polymer (GFRP/BFRP) composites under dynamic loading after seawater exposure were all reduced by seawater exposure.

Most of the existing research has focused on the effect of salt spray on FRP-bonded joints, while the related research on salt solution on FRP-bonded joints is relatively rare. It is necessary to carry out the humid and hot immersion environment tests to simulate the marine environment to provide the test data of BFRP in the marine environment. Therefore, in this paper, Araldite^®^2012 epoxy adhesive was used to study the influence of different salt solutions on the performance of BFRP joints at high temperature, the load–displacement curves of different aging times (0, 10, 20, and 30 days) were obtained, and the corresponding numerical curve of failure strength was obtained after calculation. Simple Fick’s law was used to fit the water absorption law of dumbbell-type adhesive and BFRP-based adhesive, and differential scanning calorimetry (DSC) was used to measure T_g_. Finally, the failure mode analysis of the corresponding bonding regions was carried out by macroscopic visual inspection and microscopic scanning electron microscopy (SEM) and energy dispersive X-ray spectroscopy (EDX). The change of mechanical properties of epoxy adhesives during wet–heat aging and the analysis of microscopic failure mechanism were investigated.

## 2. Materials and Methods

### 2.1. Materials

The purpose of this paper is to fabricate BFRP adhesive joints to study the effect of salt solution BFRP joint durability in high-temperature environments. A spherical epoxy adhesive was used in the test—Araldite^®^2012 (Huntsman Advanced Materials, Inc., Hong Kong, China)—which is a two-component structural epoxy adhesive (epoxy + curing agent, the ratio is 1:1). The main components are bisphenol A epoxy resin (average molecular weight < 700) 60–100% and methacrylic acid 3–7%, tough adhesive. This adhesive has very high lap shear and peel strength, can withstand large loads, and is resistant to aging, fatigue, and corrosion. It can be used to bond different substrates such as composites, metals, and thermoplastics. The board used in the test was a BFRP composite material (Jilin Zhongdao Technology Co., Ltd., Baicheng, China). The plates used in the test were BFRP composite materials processed from twill braid and basalt fiber unidirectional prepreg, which were cured at 60 °C for 4 h, cooled at room temperature, and then cured at 130 °C for 3 h. The fiber layer laying direction in BFRP board was [0/90/0/90/0/90]S, the density was 600 g/cm^3^, the epoxy resin matrix accounted for 30–35%, and the basalt fiber accounted for 65–70%. The diameter of the basalt single fiber was 11 μm, and the resin matrix was composed of two components, GT-807A (epoxy resin) and GT-807B (curing agent), and the ratio (mass ratio) was 100:20. The main performance parameters of the adhesive bond and BFRP composite board are shown in Table 1 and Table 2 (provided by the manufacturer).

### 2.2. Single-Lap Joint

BFRP single-lap joint (SLJ) was prepared in the test protocol referring to ASTM standard D5868-01 for composite–composite material joints. Bonded joints were fabricated in a dust-free environment with a temperature of (25 ± 2) °C and a humidity of (50 ± 5)%. The geometry and dimensions of the SLJ are shown in Figure 1, the overlap length and width were both 25 mm, and the total wall length was 100 mm. The surface of the BFRP joint to be bonded was wiped with acetone solution to remove dust and grease before bonding, and dried for 15 min. A special two-component glue gun was used to mix the adhesive components uniformly and then apply glue, and then glass beads with a diameter of 0.2 mm were used to control the thickness of the adhesive layer on the special bonding fixture to complete the bonding of the test pieces. The fixture is shown in Figure 2.

### 2.3. Specific Test Methods

#### 2.3.1. Experimental Design

In order to study the durability of BFRP joints with salt solution in high-temperature environment, 0, 10, 20, and 30 days were selected as aging time points. Each environment was divided into 4 groups according to the 4 experimental times, giving a total of 12 groups, and 5 specimens were selected from each group. According to the test purpose, three environmental aging durations were selected, and each immersion aging was designed based on the ISO175 standard [37].

These environments include:(i)SLJ−DI water: The single-lap BFRP joint was immersed in 80 °C/deionized water.(ii)SLJ−3.5% SS: The single-lap BFRP joint was immersed in 80 °C/3.5% NaCl solution.(iii)SLJ−5.0% SS: The single-lap BFRP joint was immersed in 80 °C/5.0% NaCl solution.

Salt solutions were used to study the effects of deicing salt solutions often found in invasive aging environments, obtained by mixing 5% by weight of sodium chloride salt with deionized water, and the salt solution concentration is defined by ASTM D1141 [38]. After the test pieces were cured, they were put into the high- and low-temperature humidity–heat alternating experimental box (WS-1000 of Weiss Equipment Experiment Co., Ltd., Dongguan, Guangdong), respectively, and the aging was carried out in the corresponding environment and time.

#### 2.3.2. Water Absorption

In order to analyze the hygroscopicity of the adhesive, a mold was specially designed to make dumbbell-shaped specimens. The two components of the adhesive were mixed and evenly filled in the mold, and a certain pressure was applied to shape it. Three specimens were fabricated using the bulk adhesive, each with a size of 150 mm × 20 mm and a thickness of 2 mm, and the geometry and dimensions are shown in Figure 3. Calculation of hygroscopicity requires periodic measurement of the mass of BFRP joints and dumbbell-shaped rubber plate specimens, and then fitting the values using simple Fick’s second law. During the weighing process of the test piece, the test piece was taken out of the test box and then wiped gently with absorbent paper to remove the surface moisture, and placed on a high-precision electronic scale with an accuracy of 0.1 mg for weight measurement. Disposable gloves should be worn throughout the weighing process and completed quickly to avoid interference from external contaminants and evaporation of water.

By measuring the mass of the adhesive at different time intervals, M_t_ can be obtained by applying the following formula:(1)$$M_{t}=\frac{w_{t}-w_{o}}{w_{o}}\times 100\%$$

The $w_{t}$ represents the mass of the sample at time t, and $w_{o}$ is the initial mass of the sample. During the measurement, in order to remove the water adhering to the surface of the sample, we gently wiped the surface of the sample with a clean, lint-free cloth.

#### 2.3.3. Tensile Test

The BFRP single-lap joint that meets the predetermined time standard was taken out of the high- and low-temperature humidity–heat alternating experimental box in time, washed with clean water, dried, and placed for 8 h before the quasi-static tensile test. A Xinguang universal testing machine was used (Jinan Xinguang Universal Testing Machine Manufacturing Co., Ltd., Jinan, China) to conduct tensile test on the joint at a constant speed of 2 mm/min. The tensile process is shown in Figure 4. To eliminate the bending stress present during the stretching process, a 2 mm-thick shim was clamped at the loading end of the joint. The specimens in each environment were tested 3 times, and the load–displacement curve was recorded to calculate the average failure strength of the joint.

#### 2.3.4. Differential Scanning Calorimetry (DSC) Analysis

The glass transition temperature (T_g_) is the temperature at which an amorphous polymer changes from a glassy state to a highly elastic state, and is an inherent property of substances. The glass transition temperature of an adhesive is a key parameter for strength and performance at a given temperature, which determines the mobility of polymer chains in the adhesive and is related to the density of the polymer chains. In order to investigate the effect of adhesive T_g_ in hot and humid environment, the thermal properties of the adhesive were measured with an instrument (Mettler Toledo, DSC3+, Switzerland). The experiment was carried out in a nitrogen environment, the ambient temperature was changed from room temperature to −50 °C, heating from −50 °C to 150 °C, holding at 150 °C for 2 min, and then repeating the above process at 150 °C. The temperature change rate was 5 °C/min. The test samples were taken from the bulk adhesive is about 5 mg, and each group of tests was repeated 3 times to obtain the average value.

#### 2.3.5. Thermogravimetric Analysis: Differential Thermo Gravimetry

Thermogravimetric analysis experiments were performed on a TAperPyris 1TGA thermogravimetric analyzer by heating an alumina pan in a N2 atmosphere at a heating rate of 10 °C/min for samples which were taken from the bulk adhesive from 25 °C to 800 °C. By recording the weight increase with temperature and weight loss rate, the instrument carried out component analysis and qualitative analysis to study the thermal stability of adhesive.

## 3. Result and Discussion

### 3.1. Hygroscopicity Analysis

Generally speaking, Fick’s second law is used to describe the unsteady diffusion process, which is usually called simple Fick’s law. The derivation process is as follows:

Due to the anisotropy of the sample material, Fick’s second law can be simplified to:(2)$$\frac{\partial C}{\partial t}=\frac{\partial }{\partial x}\left(D_{x}\frac{\partial C}{\partial x}\right)+\frac{\partial }{\partial y}\left(D_{y}\frac{\partial C}{\partial y}\right)+\frac{\partial }{\partial z}\left(D_{z}\frac{\partial C}{\partial z}\right)$$
where C is the water concentration, t is the time, D is the diffusion coefficient, and the x, y, and z axes are the spatial coordinate axes of the concentration gradient along the three directions. Let the thickness direction of the sample be the x-axis direction, because the dimensions in the other two directions are much larger than this direction, the water can be idealized as one-dimensional diffusion on the x-axis:(3)$$\frac{\partial C}{\partial t}=D\frac{\partial^{2}C}{\partial x^{2}}$$

Let the sample thickness be h(Figure 5), the initial condition is that the water concentration in the uniform sample is $C_{0}$ ($C_{0}$ = 0) when t = 0, and the two surfaces of the sample are constant water concentration $C_{\infty }$ when t ≥ 0. Combined with the diffusion Equation (2), the water concentration distribution of the sample at any position and time in the x-axis direction can be expressed as:(4)$$\frac{C(x,t)-C_{0}}{C_{\infty }-C_{0}}=1-\frac{4}{\pi }\sum_{n=0}^{\infty }\frac{{\left(-1\right)}^{n}}{\left(2n+1\right)}\exp\left[-\frac{{\left(2n+1\right)}^{2}\pi^{2}D}{h^{2}}t\right]\cos\frac{(2n+1)\pi x}{h}$$

The integral of the water concentration at each x position is the water absorption of the sample at time t:(5)$$M_{t}=\int_{-h/2}^{h/2}\left(C-C_{0}\right)dx=\int_{-h/2}^{h/2}Cdx$$

After integrating (4), we can obtain:(6)$$\frac{M_{t}}{M_{\infty }}=1-\frac{8}{\pi^{2}}\sum_{n=0}^{\infty }\frac{1}{{\left(2n+1\right)}^{2}}\exp\left[-\frac{{\left(2n+1\right)}^{2}\pi^{2}D}{h^{2}}t\right]$$

The diffusion coefficient D can be obtained from the following formula:(7)$$D=\frac{\pi }{16}{\left(\frac{h}{M_{\infty }}\right)}^{2}{\left[\frac{M(t_{1})-M(t_{2})}{\sqrt{t_{1}}-\sqrt{t_{2}}}\right]}^{2}$$

Moisture diffuses into the polymer matrix in different ways (including moisture entry rate, moisture distribution through thickness, and moisture absorption capacity), depending on several molecular and microstructural aspects, such as molecular structure polarity and cross-linking [39] degree. During the absorption process, two water molecules, bound water and free water, exist in the epoxy polymer matrix. Free water will occupy the free space of the polymer, causing plasticization of the material. On the other hand, the bound water forms hydrogen bonds with the epoxy network, causing the material to expand, plasticize, and the reduce strength and the glass transition temperature [34].

According to the diffusion coefficients presented in Table 3, MATLAB software was used to draw the comparison between the FICK fitting curve and the experimental curve in each environment, as shown in Figure 6. The fitted curve grows relatively fast at the beginning, and then gradually slows down and reaches saturation. The basic trend is consistent with the experimental data, which can be roughly divided into two stages. In the first stage, the type I bound water should play a dominant role in the water absorption process of the adhesive, destroy the van der Waals force between the initial chain and the hydrogen bond, enhance the mobility and swelling of the polymer segment, and lead to the plasticization of the epoxy resin. The second stage should be that type II bound water increases with exposure time, slowing down the effect of type I bound water. Type II bound water will not plasticize the epoxy resin, but it can enable secondary cross-linking with hydrophilic groups, such as hydroxyl and amine in the polymer chain network, weaken the stress between the polymer molecular chains, and lead to irreversible material changes [17]. According to Table 3, from the perspective of the overall water absorption process, the water absorption capacity of the adhesive and BFRP in the salt solution environment is lower than that in the deionized water environment. According to Jones [39], the cross-linked matrix acts as a semi-permeable membrane where water is permeable but large inorganic ions are hindered. This effect stems from the cross-linking behavior of the polymer, which may act as a semi-permeable membrane, enabling the movement of water and stopping large inorganic ions [27], resulting in lower water uptake by specimens in saline solution environments. It can also be found that as the concentration of the salt solution increases, the water absorption gradually decreases. This may be because the greater the concentration of water molecules outside the adhesive, the faster the water molecules enter the free space in the adhesive, and the greater the water absorption rate. At the same time, according to the inference in the following paragraph that the joint is more destructive when soaked in deionized water than salt solution, Mohsen Heshmati’s [27] theory is confirmed, that the influence of water molecule content on the property of the joint at the same temperature is related to the higher near-surface water concentration of the sample immersed in deionized water. Therefore, the authors believe that the activity of the salt solution is weaker than that of deionized water, resulting in a decrease in the saturation of the water in the test piece.

In addition, in the many starting points of the first stage, it was found that there was a large gap between the experimental data and the fitted data in the two figures. The reason for this phenomenon may be that the sample jumps from the air humidity of 50 ± 5% to the solution humidity of 100%, while the fitting data has a gradual increase, so the experimental data are slightly larger than the corresponding data. At the same time, it can be found that the saturated water absorption of adhesives in the three environments is much higher than that of BFRP, mainly because of the presence of basalt fibers with 65–70% volume fraction of no or little water absorption features in BFRP. It is worth mentioning that the mass decrease occurs in the mass increase process of the joint absorbing water, which is due to the loss of low molecular components in the water absorption process, and this loss degree is expected to increase with the increase in temperature. However, the degradation of water and temperature is considered to be the most likely factor [34], because water easily infiltrates the polymer polarity changes the properties of polymer materials of ontology, and temperature accelerates the diffusion process and makes water enter the water absorption and degradation mechanism, resulting in intensified degradation. The quality of low-molecular components has little influence on the process of increase in water quality.

### 3.2. Tensile Test Analysis

Tensile shear testing is the traditional method for testing the mechanical properties of materials, and by this method, the ability of adhesives to be used in BFRP joint structural applications was investigated. There are several types of tests available for this purpose, but the tensile shear test was chosen because this method of providing mechanical properties of bonded joints is widely accepted.

Figure 7a is the most representative load–displacement curve of different aging cycles obtained by single-lap joints using Araldite^®^2012 adhesive in an 80 °C/DI water environment. After 10 days of aging, it can be seen that the joint failure load was significantly reduced, and the stiffness is slightly increased. The load–displacement curve did not change significantly after aging for 20 days, indicating that the joint degradation progressed slowly during the aging process of 10–20 days. After 30 days of aging, it was observed that both the failure load and the fracture displacement reached the minimum value in the experimental period. There is no doubt that this time period is the most serious for joint degradation. At this time, the stiffness of the joint was still slightly higher than that of the unaged group. Figure 7b,c are the most representative load–displacement curves of different aging cycles obtained by single-lap joints using Araldite^®^2012 adhesive in 80 °C/3.5% SS and 80 °C/5.0% SS environments. After 30 days of aging, the performance of the joint deteriorated significantly. Compared with the unaged conditions, the failure load decreased from 9.33 kN to 7.10 KN and 6.34 KN. The maximum displacement of the fracture decreased from 3.96 mm to 2.15 mm and 1.74 mm, and the stiffness also increased slightly. In the 80 °C/5.0% SS environment, the joint failure load decreased obviously after 30 days of aging, corresponding to the joint failure load after 20 days of aging in 80 °C/3.5% SS environment and the joint failure load after 10 days of aging in the 80 °C/DI water environment. It shows that the order of joint degradation speed under the three environments is the largest at 80 °C/DI water, followed by 80 °C/3.5% SS, and the smallest at 80 °C/3.5% SS. The concentration of salt solution delays the hydrolysis of bonding. It is inferred that the salt solution can delay the hydrolysis of the adhesive, and the higher the concentration, the stronger the delaying performance of the bonded joint. Regarding the failure displacement of the adhesive layer, as shown in Figure 7, it can be seen that when the adhesive layer is broken, the load–displacement relationship of the joint has the same evolution trend. That is, with the increase in aging time, the failure load decreased, and the failure displacement had the same decreasing trend. This was due to joint aging corrosion that stimulated a reduction in the effective area of bond lines, leading to premature fracture at relatively low loading levels [40]. In addition, it can be clearly seen that the stiffness of the joint showed an increasing trend under different aging conditions. This trend was mainly due to the fact that the joints were cured at room temperature during the preparation process, and the joints were further exposed to a hot and humid environment at 80 °C for a post-cured effect. Adhesives affected by high temperature increase the crosslinking of polymer chains, which changes the crosslinking density of the resin [41], resulting in an increase in the stiffness of the joint.

As shown in Figure 8, for Araldite^®^2012 joints at 80 °C/DI water, the average failure strength decreased by 18.31%, 27.98%, and 38.63% after 10, 20, and 30 days of aging, respectively. In the 80 °C/3.5% SS environment, the reductions were 7.15%, 23.11%, and 31.16%. In the 80 °C/5.0% SS environment, the reductions were 11.45%, 18.31%, and 30.41%. It can be clearly seen that the average failure strength of the joints in the three environments showed an overall downward trend. The main reason for this trend is that in a humid and hot environment, moisture tends to easily penetrate into FRPs through the adhesive joints, and the interface between the epoxy resin and the matrix resin deteriorates with time, resulting in a decrease in the mechanical properties of the joint. It can also be observed that, in the given aging conditions, the strength of the joints soaked in deionized water was lower than the average failure strength of the joints of the two salt solutions. So, it can be inferred that immersion of the joint in deionized water environment was more destructive than saline solution environment. In Figure 8 it can be observed that the failure strength of 80 °C/3.5% SS and 80 °C/5.0% SS varied in different periods during the aging process, mainly due to the infiltration effect of moisture [42]. In the deionized water environment, when the water was sucked into the adhesive, it formed an electrolyte with the water-soluble substances in the adhesive, and the water was driven into the adhesive under the action of osmotic pressure to balance the concentration difference. However, in the salt solution environment, the high concentration of the NaCl solution leads to reverse osmosis, and water diffuses out from the binder, balancing the concentration difference between the binder and the environment, so that the water absorption of the binder decreased with the increase in the concentration of salt ions. Salt ions speed up water absorption, so it takes less time to reach dynamic equilibrium at 5%.

### 3.3. Differential Scanning Calorimetry Analysis

T_g_ is a key parameter to control the mechanical properties of adhesive and epoxy matrix BFRP materials at different temperatures. Differential scanning calorimeter was used to analyze Araldite^®^2012 adhesive at 80 °C/DI water, 80 °C/3.5% SS, and 80 °C/5% SS. The results are shown in Figure 9. It can be seen in Figure 9 that the glass transition temperature after aging was lower in all three environments than in the unaged one. When Araldite^®^2012 was not aged, the T_g_ was 44.24 °C, and when aged at 80 °C/DI water for 10 days, 20 days, and 30 days, the T_g_ was 29.73 °C, 25.51 °C, and 21.49 °C, with a decrease of 32.80%, 42.34%, and 51.42%, respectively; after aging at 80 °C/3.5% SS for 10 days, 20 days, and 30 days, the T_g_ was 27.37 °C, 24.47 °C, and 23.99 °C, with a decrease of 38.13%, 44.69%, and 45.77%, respectively; after aging at 80 °C/5% SS for 10 days, 20 days, and 30 days, the T_g_ were 28.37 °C, 26.62 °C, and 24.23 °C, with a decrease of 35.87%, 39.83%, and 45.23%, respectively. On the whole, the polymer molecular chain of the adhesive was broken in the high-temperature and high-humidity environment, which increased the mobility of its molecular chain, resulting in the expansion, plasticization, cracking, and hydrolysis of the adhesive [24]. In this work, it can be considered that moisture is mainly absorbed as bound water. A lot of type I bound water was created when the water penetrated into the adhesive. They acted as a plasticizer, disrupting the van der Waals initial force between the polymer chains and hydrogen bonds, resulting in increased segment mobility and lower T_g_ [17].

### 3.4. Thermogravimetric Analysis: Differential Thermo Gravimetry

A detailed TGA analysis was performed on Araldite^®^2012. The results of thermal degradation related data for the samples are presented in Figure 10. Table 4 lists the temperature IPT when the initial mass decays caused by heating (this parameter is used for the thermal stability of the tested material), the maximum temperature, Tmax, when the degradation rate was the largest, and the sample residual rate.

According to the TG and DTG curves in Figure 10 and Table 4, after the adhesive underwent three environmental aging treatments, the TG curves of different aging times decreased in steps and almost overlapped, and the weight loss was mainly concentrated at 340–450 °C. In the region of 25–339 °C, the reduction in weight loss (<4 wt.%) was very small, corresponding to water evaporation. In the region of 340–450 °C, the DTG curves of different aging times had only one peak, indicating that only one degradation occurred, and the quality loss ranged from 86% to 91%. It is the complete decomposition of epoxy resin and some organic compounds (including methacrylic acid and its hardener). Specifically, during the heating process, the heat gradually penetrated into the epoxy resin molecules, causing all the molecules to decompose in a short time [43]. When the temperature was raised to 800 °C, about 5% by weight of residue remained, mainly related to inorganic compounds (as basalt fibers in the form of solid residues and basalt fiber mixed with C). The maximum decomposition rate of the aged samples was generally greater than that of the unaged samples, and the maximum degradation rate was basically around 372 °C. IDT is used to describe the thermal stability of Araldite^®^2012. As mentioned earlier, the higher the temperature, the higher the thermal stability of the sample. The IPT of the aged samples was around 345 °C and decreased slightly with the prolongation of the aging time. The ITP decreased but the trend fluctuation was small, indicating that there was no significant difference between the thermal stability of the aged viscose and the unaged viscose.

### 3.5. Analysis of Failure Section Morphology

In order to study the failure modes of BFRP joints in three immersion environments, the failure mechanism of fracture properties was analyzed. Figure 11 shows the representative fracture surface appearance of BFRP joints in each environment.

The failure modes of a single joint include (1) tear failure, (2) cohesive failure, and (3) interfacial failure. However, in real life, there is no single failure mode in the bonding overlap area of most SLJ specimens, and the specimens usually exhibit mixed failure modes, involving a combination of the above two or even three modes.

In this case, we judged the failure mode of the joint by visually inspecting the size and combination of the bond area of each failure mode in typical failure modes. The exposed fiber filaments were mainly visible in the failure section of the aged joint. It was easy to judge that there was a tear failure mode in the joint only by visually inspecting the exposed fiber filaments in the bonding area. However, according to the microscopic analysis in Figure 11, it was found that the failure mode of the bonding area after 30 days of aging was interface failure. The appearance of this phenomenon may be caused by the adhesion of a layer of fibrous cloth with poor adsorption on the surface of BFRP to the adhesive. The failure sections of the unaged joints were all tear failures, indicating that the bonding process gave full play to the bonding properties of the adhesive. With the increase in aging time, the tear failure area of the failure section decreased significantly, indicating that the mechanical properties of the joint were significantly reduced, and the cohesive failure was a relatively weak manifestation of the decline of the mechanical properties of the joint. After 10 days of aging in deionized water, a large area of cohesive failure occurred on the failure section. At this time, the cohesive strength of the BFRP composite was the same as that of the adhesive and the interfacial cohesive strength of the joint. After aging at 80 °C/3.5% SS and 80 °C/5.0% SS for 10 days, the tear failure area of the failure section did not decrease significantly. However, after aging in deionized water for 20 days, the tearing failure area of the failure section was significantly reduced, and an interfacial failure occurred. In the deionized water environment, the fiber tear area of the failure section decreased the fastest, indicating that the mechanical properties of the joint decreased faster in the deionized water environment. The failure mode of the joint shows a transition from tearing failure to cohesive interface hybrid failure with the increase in time. At the same time, according to the analysis of the moisture absorption rate above, it was found that the degree of decline in the mechanical properties of the adhesive joint was related to the amount of water absorbed by the adhesive. This was because, in the process of damp heat aging, water molecules penetrate into the interface of the composite material through the bonding seam, and the water absorption of the adhesive causes the expansion of the resin and the fiber matrix to generate expansion stress, which reduces the mechanical properties of the adhesive joint.

### 3.6. SEM-EDX Analysis

Scanning electron microscopy was performed on selected failure areas to study fracture characteristics. Figure 12 compares the SEM microstructure of the joint section before and after the aging environment, and analyzes the failure mechanisms involved through the characteristic section. Specifically, in Figure 12(a2), there is fiber damage, and the reason for the fiber damage should be formed during the bonding and the stretching of the composite material, indicating that the interface between the fiber and the resin failed seriously during the stretching process. In Figure 12(c1), there are some voids in the BFRP resin, and the reason for the voids should be that the voids in the resin matrix were not completely eliminated during the fabrication of the composite material. As one of the matrix materials, resin plays an important role in fiber load transfer in composites. The presence of voids may degrade the mechanical properties of the material, especially the matrix-dominated properties, such as the interlaminar shear strength and flexural properties [44]. There are cracks in Figure 12(c2), and the reason for the cracks is that the chemical degradation of the polymer led to the increase in pores inside the entangled polymer chains, which promoted chain expansion and eventually formed microcracks in the polymer network [45]. A large number of matrix fragments can be seen on the surface of the joint after fracture in an aging environment of 0–20 days. In contrast, relatively few matrix fragments can be observed on the surface of the basalt fiber after 30 days of aging in either saline solution or deionized water, as shown in Figure 12(c4). This is mainly due to the fact that water diffuses into the adhesive through the bulk adhesive, along the interface, capillary, and porous adhesive of the composite material, changing the crosslinking of the polymer and increasing the binding force between the adhesive and the resin matrix [23]. At the same time, we can also observe that the degree of interfacial debonding between the fiber and the resin matrix was higher over time. At the same time, we can also observe that the degree of interfacial peeling between the fiber and the resin matrix was higher over time, which indicated that the bonding between the fiber and the matrix interface was significantly damaged by moisture, and there was obvious interfacial peeling between the fiber and the resin matrix.

In order to further explore the fracture characteristics, the EDX method was used to analyze the element distribution on the surface of the failure area. From Figure 13, the exposed vertical layered fibers and resin matrix can be clearly observed. Analysis of the C and O element maps and data showed that the concentration of these two elements was relatively high, the content of C element was close to 76%, and the content of O element was close to 21%, and the two added up to nearly 98%. The trace of O element was obviously distributed along the fiber, while the distribution position of C element was just opposite to that of O element. There was almost no C element in the fiber exposure, indicating that O element mainly existed in the fiber, while C element mainly existed in the epoxy resin matrix. From Table 5, in the 80 °C/DI water environment, after 10 days and 30 days of aging treatment, the carbon element content was 79.14% and 76.97%, the oxygen element content was 19.44% and 19.56%, and the O/C values were 0.24 and 0.25, respectively; in the 80 °C/3.5% SS environment, after 10 days and 30 days of aging treatment, the carbon content was 76.17% and 74.65%, the oxygen content was 21.61% and 21.75%, and the O/C value was 0.28 and 0.29, respectively; in the 80 °C/5.0% SS environment, after 10 days and 30 days of aging treatment, the carbon content was 75.17% and 73.93%, the oxygen content was 21.76% and 23.93%, and the O/C value was 28.95 and 32.37, respectively. The hydrolysis of the adhesive at high temperature increased the O element on the failure surface, so the change of the O/C value reflected the degree of hydrolysis of the material. It can be seen that with the increase in aging time in the aging process, the degree of hydrolysis of the adhesive was higher. It is worth mentioning that it is easy to know that Si is an element that does not exist in adhesives and epoxy resins, and one of the main components of basalt fiber is SiO_2_, so Si can be used as a characteristic element of basalt fiber. By analyzing the content of Si, the degree of exposure or damage of basalt fibers can be obtained, and it was found that the content of Si in the experimental data was consistent with the failure mode analysis of the previous joint.

## 4. Conclusions

In this paper, experimental tests were carried out on BFRP single-lap joints to study the effects of immersion in three aging environments on mechanical properties of bonded joints. In this paper, the mechanical properties of the joints were measured by the quasi-static tensile shear method; the hygroscopic properties of the joints were analyzed by the simple Fick model; the glass transition temperature (T_g_) of the adhesive was measured by DSC; and the thermal stability of the adhesive was evaluated by the TGA-DTG curve. Finally, the surface conditions and failure mechanism of the substrate were characterized by surface macroscopic and SEM-EDX microscopic analysis. Under the condition of known test results, the following conclusions can be drawn:(1)The high-temperature environment increased the crosslinking of the polymer chain in the adhesive and changed the crosslinking density of the resin, leading to the curing reaction with increased stiffness after the occurrence of the adhesive.(2)The joint did not show more adverse effects in salt solution immersion than deionized water, which was mainly related to the cross-linking infiltration effect of the polymer. The formation of a semi-permeable membrane stops the movement of water and large inorganic ions forming osmotic pressure difference, which makes the activity of the salt solution weaker than the deionized water and reduces the water absorption saturation of specimens.(3)Under a high-temperature environment, the adhesive broke the polymer molecular chain, which increased the mobility of its molecular chain, and caused the adhesive to expand, plasticize, crack, and hydrolyze. There was no significant difference in the thermal stability of the adhesive before and after aging, and 340–450 °C was the main weight loss stage, where primary degradation occured, which is the complete degradation of epoxy resin and some organic substances.(4)In the analysis of joint failure, through macroscopic visual observation of the size of the bonding area of each failure mode, it was found that the bonding area presents a trend of transition from tearing failure to late cohesion interface hybrid failure as time goes on. Through microscopic SEM-EDX image analysis, with the increase in aging time, the degree of adhesive hydrolysis was higher, the bond between fiber and matrix interface was significantly damaged by moisture, and the degree of interface peeling was higher.

## Figures and Tables

**Figure 1 polymers-14-02250-f001:**
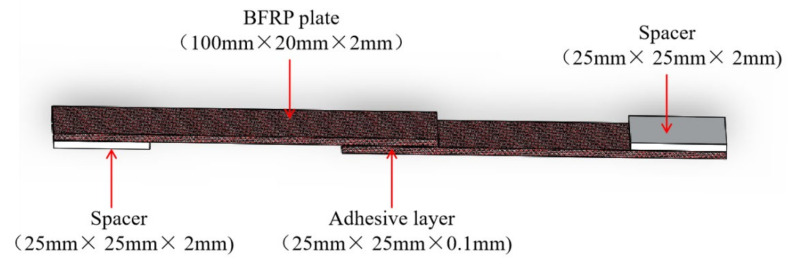
Geometry of single-lap adhesive joint.

**Figure 2 polymers-14-02250-f002:**
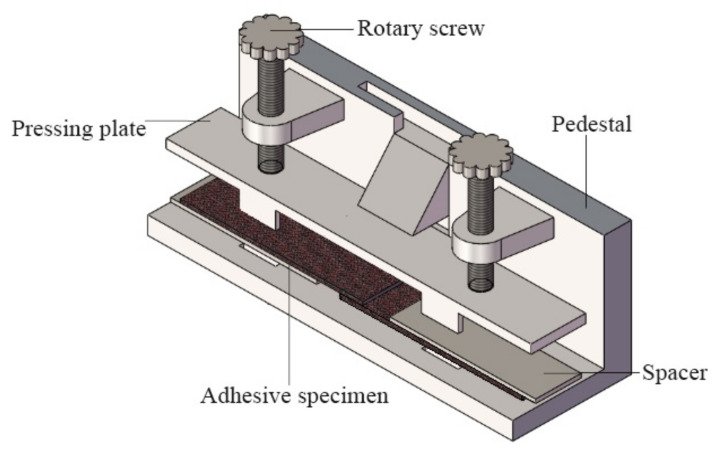
Adhesive fixture of single-lap joints.

**Figure 3 polymers-14-02250-f003:**
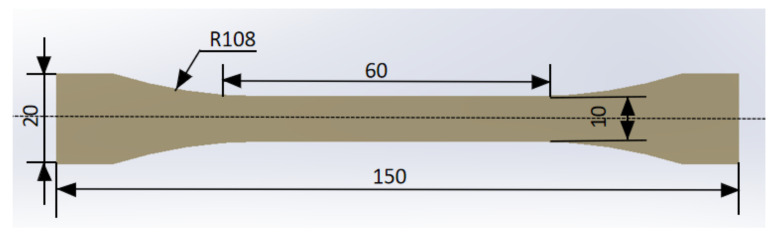
Geometric shape of dumbbell shaped specimen.

**Figure 4 polymers-14-02250-f004:**
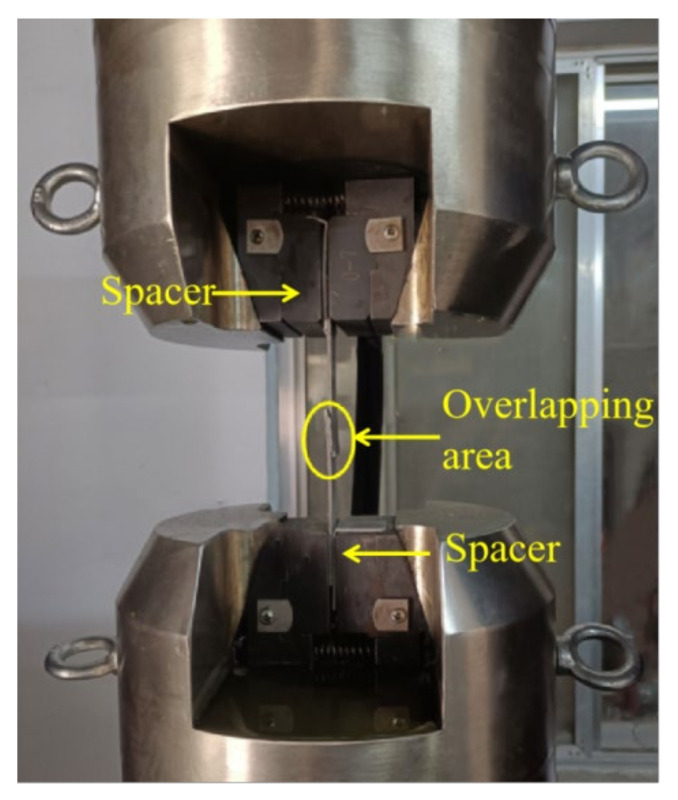
Quasi-static state tensile test site.

**Figure 5 polymers-14-02250-f005:**
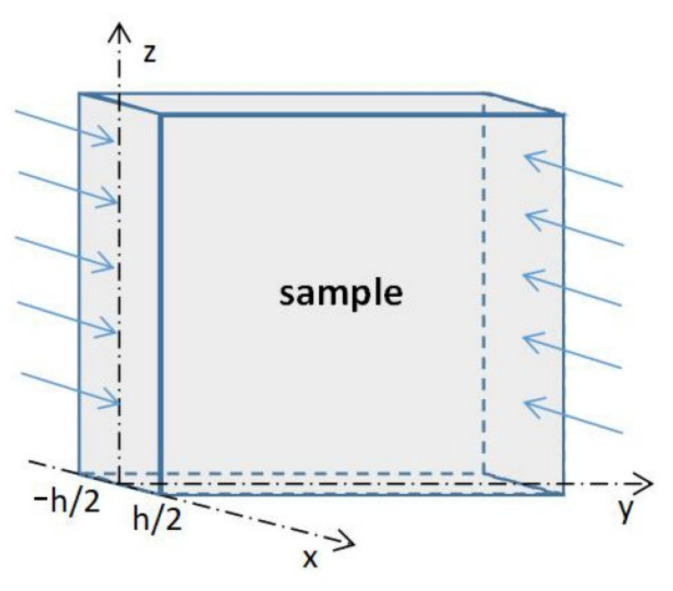
Diffusion in large plane sheet sample space.

**Figure 6 polymers-14-02250-f006:**
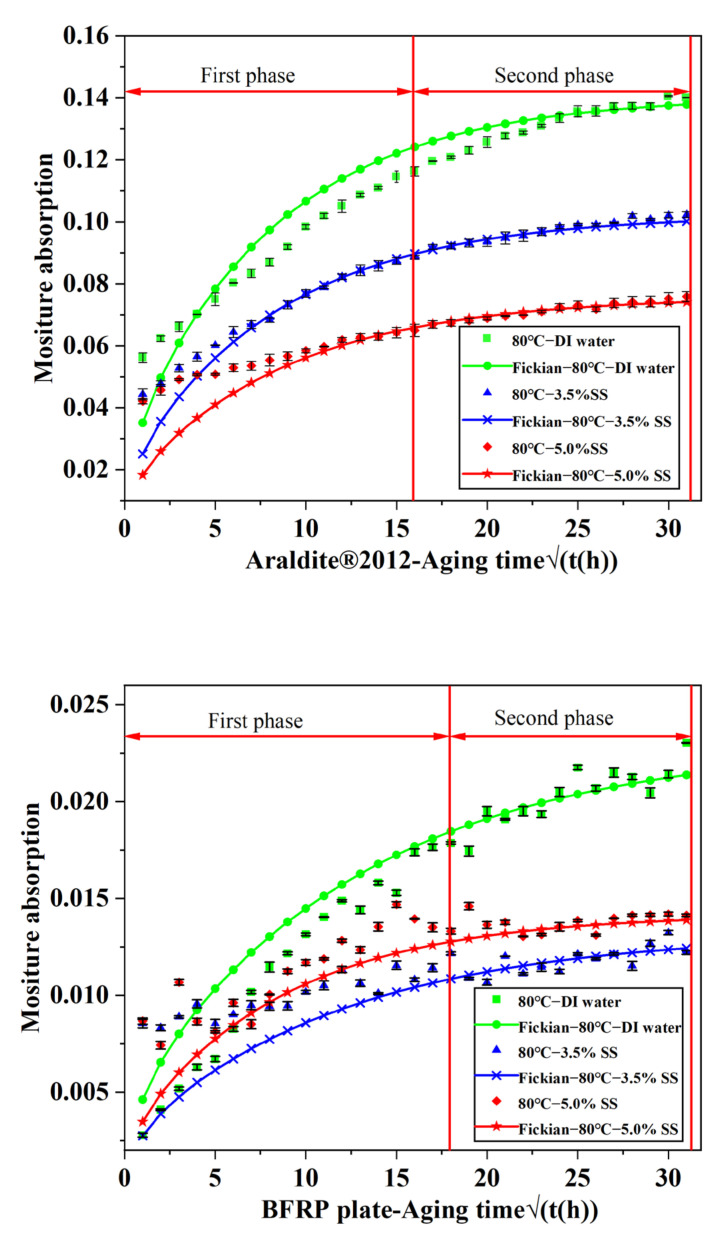
The moisture absorption of BFRP and Araldite^®^2012.

**Figure 7 polymers-14-02250-f007:**
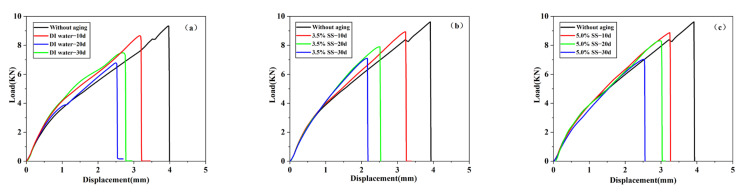
The most representative load displacement curves of different aging stages in different environment were obtained: (**a**) SLJ-DI water; (**b**) SLJ-3.5% SS; (**c**) SLJ-5.0% SS.

**Figure 8 polymers-14-02250-f008:**
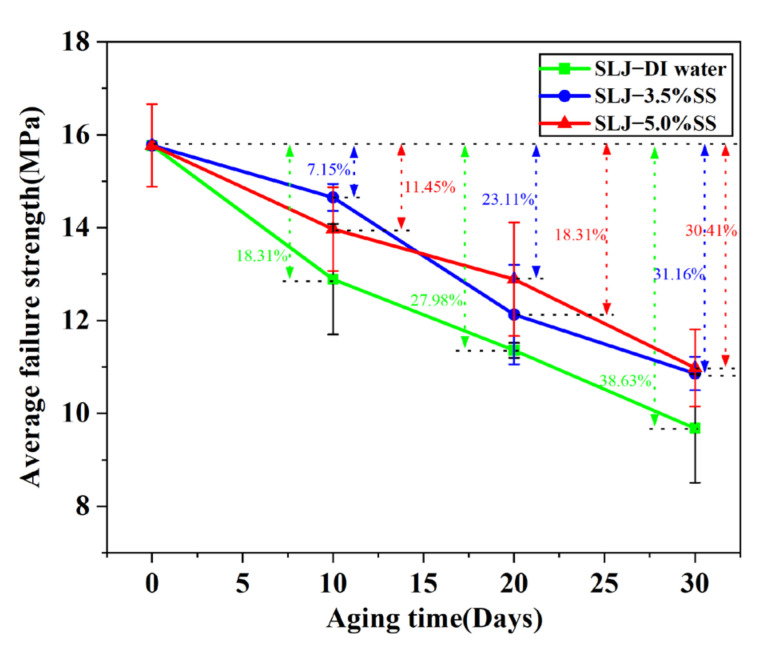
Effect of hygrothermal aging on the strength of single-lap joint.

**Figure 9 polymers-14-02250-f009:**
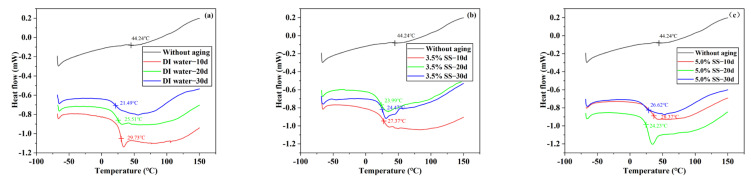
DSC thermogram of Araldite^®^2012 adhesive in different environments: (**a**) DI water; (**b**) 3.5% SS; (**c**) 5.0% SS.

**Figure 10 polymers-14-02250-f010:**
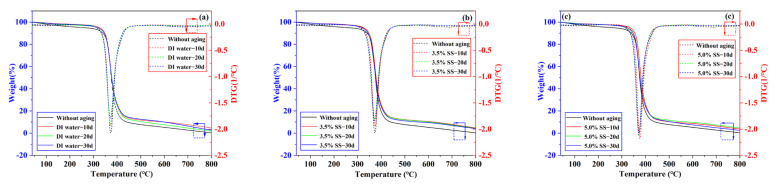
TGA and DTG thermogram of Araldite^®^2012 adhesive in different environments: (**a**) DI water; (**b**) 3.5% SS; (**c**) 5.0% SS.

**Figure 11 polymers-14-02250-f011:**
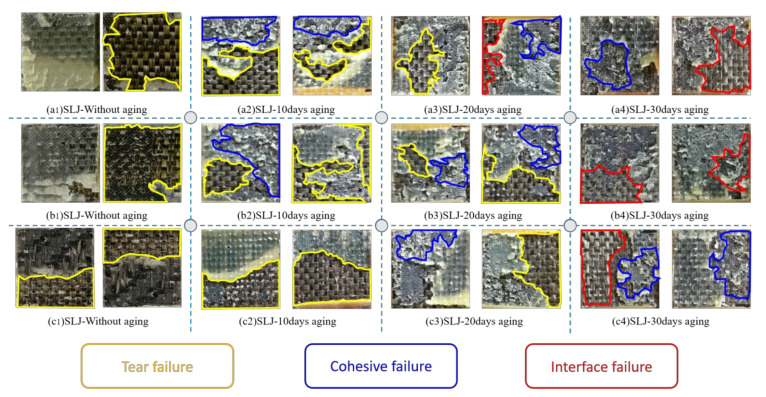
(**a****1**–**a4**) The surface of Araldite^®^2012 joints in DI water; (**b****1**–**b4**) the surface of Araldite^®^2012 joints in 3.5% SS; (**c****1**–**c4**) the surface of Araldite^®^2012 joints in 5.0% SS.

**Figure 12 polymers-14-02250-f012:**
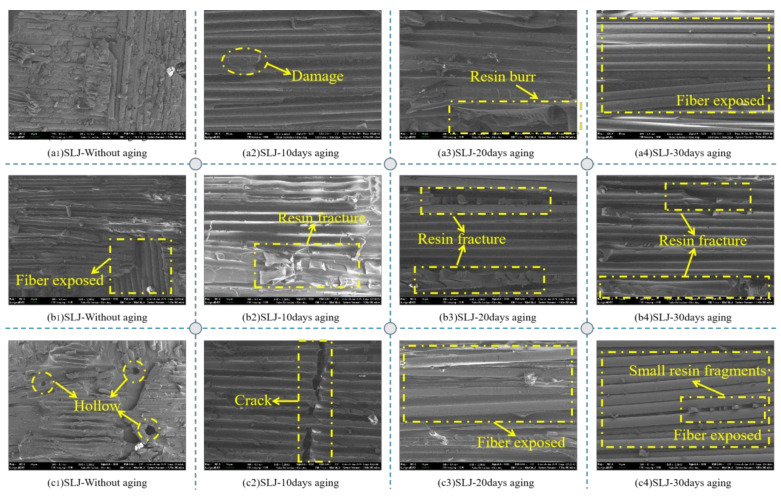
The SEM micrograph of different adhesive joints in different environments: (**a****1**–**a4**) the surface of Araldite^®^2012 joints in DI water; (**b****1**–**b4**) the surface of Araldite^®^2012 joints in 3.5% SS; (**c****1**–**c4**) the surface of Araldite^®^2012 joints in 5.0% SS.

**Figure 13 polymers-14-02250-f013:**
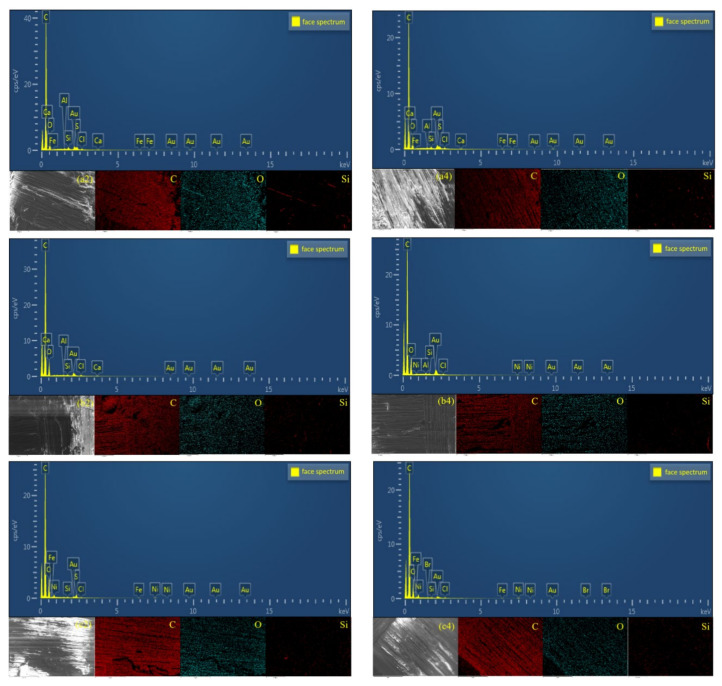
The EDX image of Araldite^®^2012 adhesive joints in different environments: (**a2**/**a4**) aging 10/30 days in DI water; (**b2**/**b4**) aging 10/30 days in 3.5% SS; (**c2**/**c4**) aging 10/30 days in 5.0% SS.

**Table 1 polymers-14-02250-t001:** Material properties of Araldite^®^2012.

	Araldite^®^2012
Young’s modulus, E [GPa]	1.65
Shear modulus, G [GPa]	0.25
Density/(kg/m^3^)	1.18
Poisson’s radio	0.43

**Table 2 polymers-14-02250-t002:** Material properties of BFRP.

Surface Density/(g/cm^3^)	Tensile Strength/MPa	Young’s Modulus/(MPa)	Nominal Thickness/mm	Single Fiber Size/μm
600	2100	105	0.115	11

**Table 3 polymers-14-02250-t003:** The moisture absorption of BFRP and Araldite^®^2012.

Samples	Condition	Saturation Moisture Uptake Mm (%)	Diffusion Coefficient D × 10^−3^ (mm^2^/h)	Thickness (T = 2 h, mm)
Araldite^®^2012	DI water	14.05%	2.05	2
	3.5% SS	10.23%	1.98	2
	5.0% SS	7.59%	1.92	2
BFRP	DI water	2.30%	1.32	2
	3.5% SS	1.32%	1.61	2
	5.0% SS	1.42%	1.96	2

**Table 4 polymers-14-02250-t004:** TGA-DTG data for Araldite^®^2012.

Environment	Aging Time	IPT (°C)	Tmax (°C)	Residue Rate (%)
Without aging	0 days	340.0	373.33	0.34%
DI water	Aging 10 days	348.7	372.5	4.89%
	Aging 20 days	346.2	371.7	1.88%
	Aging 30 days	345.7	371.7	2.68%
3.5% SS	Aging 10 days	347.5	374.2	4.82%
	Aging 20 days	343.3	371.2	4.43%
	Aging 30 days	344.2	372.5	4.15%
5.0% SS	Aging 10 days	355.3	375.8	3.79%
	Aging 20 days	348.8	371.7	4.82%
	Aging 30 days	347.9	371.7	2.56%

**Table 5 polymers-14-02250-t005:** Different element content in EDX.

Environment	Aging Time	C (Element)	O (Element)	Si (Element)
80 °C-DI water	Aging 10 days	79.14	19.44	0.27
	Aging 30 days	76.97	19.56	0.23
80 °C-3.5% SS	Aging 10 days	76.17	21.61	0.09
	Aging 30 days	74.65	21.75	0.08
80 °C-5.0% SS	Aging 10 days	75.17	21.76	0.09
	Aging 30 days	73.93	23.93	0.07

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
