# Peer review of "Study on the Effect of Salt Solution on Durability of Basalt-Fiber-Reinforced Polymer Joints in High-Temperature Environment"

_polymers, 2022, doi:10.3390/polym14112250_

Round 1
Reviewer 1 Report
Some minor comments are listed as follows.
Figs. 7, 9, 10, please put two images in one line.
Line 569, what is mean of "thcal properties oe residual mechanical properties"?
Line 570, "mechanic"?
In the figures, change "Mpa" to "MPa".
Please try to change load to stress in analysis in for example Fig. 7.
The reference format does not fit the journal. Please improve.
An important reference with a similar topic "Degradation of the in-plane shear modulus of structural BFRP laminates due to high temperature" is ignored by the authors when doing introduction.
Author Response
Thank you very much for affirmation of our work. The reviewer's comments are of great help to our current manuscript. The point to point responds to the reviewer’s comments are listed as following.

Reviewer 2 Report
The current paper systematically studies the performance evolution of basalt fiber reinforced polymer joints immersed in salt solution at high temperature. Water absorption, mechanical properties and microscopic analysis were carried out to obtain the long-term performance evolution mechanism. Overall, the paper is systematic and well organized. Some further improvements still need to be made to clarify some information.
*Abstract
1) In this part, the authors should give some relevant mechanism explanation and analysis. For example, why the failure strength of the joint decreased slightly in salt solution environment compared to deionized water? Is there any chemical reaction or degradation that leads to the above phenomenon?
2) In addition, how to evaluate the effect of post curing reaction and degradation reaction on long-term performance of BFRP during the exposure?
*Introduction
1) Without the advantage comparison with the others’ FRPs, the introduction of FRP performance and advantages (Line 63-65) is not rigorous and accurate. For CFRP, it has very excellent mechanical properties, fatigue/creep/corrosion resistances, but carbon fiber has a high price. For GFRP, its mechanical properties are relatively good and has low cost. However, its long-term performance, especially exposed to alkaline environment is vulnerable due to the chemical degradation. Through comparing the above information, the advantages and performance of BFRP should be summarized and analyzed carefully. Please see latest research achievements of CFRP, GFRP and BFRP. Composite Structures, 2022. 281: 115060. Polymers, 2021, 13(21): 3721. Corrosion Science 138 (2018) 200–218.
2) In the present paper, the maximum immersion time is only 30 days. Is this enough to cause performance degradation for BFRP joints? In addition, the time interval is only 10 days, which is also relatively short.
3) Why do you use salt solution and deionized water as the exposure medium? What actual situation are simulated through the above medium? Please provide a relevant explanation.
*Materials and Methods
1) Please provide the curing conditions of adhesive and BFRP. In addition, what is the production process of BFRP? What is the volume fraction of fiber and matrix?
2) For part 2.3.2, generally, the water absorption test of polymer or composite materials adopts the rectangular shape. Furthermore, the water absorption and diffusion behavior along the thickness direction can be described and analyzed by Fick diffusion law (Figure 5). For the present dumbbell shaped specimen (irregular shape), whether the water absorption and diffusion behavior of the material comply with Fick’s law, please provide relevant basis and explanation.
3) During the tensile test of BFRP joints, why not use extensometer or strain gauge to obtain the strain or displacement? As known, the displacement of the collet of the tension machine is unreliable due to the slip. In addition, the bond-slip curve of BFRP joints can be further obtained by arranging a certain strain gauge.
4) Please indicate the sample preparation method and test conditions (before and after the aging) during the test of DSC and TGA.
*Result and discussion
1) Please check the superscript writing for diffusion coefficient. The title of Table 3 and table 4 are inappropriate. They should be water absorption properties, not thermal and mechanical properties. In addition, it should be further clarified that one is adhesive and the other is BFRP, and the title should correspond to figure 6.
2) What causes such a big gap of saturation moisture uptake for adhesives and BFRP in Table 3 and 4?
3) Please provide the standard deviation of moisture absorption for 80oC-DI water environment in Figure 6. In addition, how to distinguish the first stage and the second stage of water absorption? Please provide relevant basis and explanation.
4) With the increase of salt solution concentration, the gradual decline of water absorption is observed. Please provide relevant explanations.
5) Page 11, line 300-308, the authors claim type I bound water formed in the first stage, type III bound water formed in the second stage. Generally speaking, the types of water molecules inside polymer and composites can be determined by desorption tests through the drying at different temperatures as shown in Polymer testing, 2020; 90: 106761. This is because different levels of water molecules will be removed from resins and composites at different temperatures. Therefore, it is abrupt to explain the types of water molecules during the long-term exposure without the relevant desorption tests.
6) In figure 7, if the displacement is obtained through the collet of tensile machine, it is meaningless. This is because there will be a certain slip phenomenon between the anchorage and the sample during the tensile process.
7) The Mpa in Figure 8 is incorrectly written, it should be MPa. In addition, it can be found that the three exposure conditions have little effect on the average strength of single lap joint. Is this related to the shorter aging time?
8) As can be seen from Figure 9, the glass transition temperature of adhesive after aging is reduced to below 30 ℃. This means that the state of high elasticity occurred in the general summer environment. Therefore, how is it used in real practical engineering applications? Why not choose an adhesive with a higher glass transition temperature?
9) The conclusion should be further condensed, including only 3 ~ 4 key points and stating some key results and findings.
Author Response

(The authors gave the same response as above.)

Round 2
Reviewer 2 Report
The author has replied to all the comments of the reviewers, and the paper meets the requirements for the publication. It is suggested to accept the current paper.